# Residual-aided CSI-free end-to-end learning for multiuser MIMO

Emmanuel Ampoma Affum[1]*, Osumanu Futa[2], Maxwell Afriyie Oppong[3], Daniel Owusu Biney[4]

1 Faculty of Electrical and Computer Engineering, Kwame Nkrumah University of Science and Technology, Kumasi, Ashanti Region, Ghana, 2 Department of Electrical and Electronic Engineering, Accra Technical University, Accra, Greater Accra Region, Ghana, 3 Department of Electrical and Electronic Engineering, Kumasi Technical University, Kumasi, Ashanti Region, Ghana, 4 Faculty of Computer and Electronic Engineering, Kwame Nkrumah University of Science and Technology, Kumasi, Ashanti Region, Ghana

* ampoma.uestc@yahoo.com

## Abstract

A paradigm shift from Channel State Information (CSI)-dependent architectures to intelligent, AI-native air interfaces is required as 6G wireless systems advance. Conventional Multi-User Multiple-Input Multiple-Output (MU-MIMO) systems have substantial pilot overhead and computational complexity since they rely on explicit CSI for beamforming and interference management. This study suggests a novel **Deep Unfolding Successive Over-Relaxation (DU-SOR)** paradigm to overcome these constraints. In contrast to conventional end-to-end learning techniques that operate as "black boxes," DU-SOR combines iterative residual refining with a sparse Graph Transformer. The network can intuitively solve the inverse problem without explicit channel matrix inversion thanks to this novel architecture, which uses graph priors to condition the signal estimation. Extensive empirical analyses show that the proposed framework accomplishes three main goals: (i) near-optimal performance, confirmed by a mutual information score of 0.98 at 20 dB SNR; (ii) mathematically proven scalable complexity, reducing the scaling order from $\mathcal{O}(K^3)$ to $\mathcal{O}(K \log K)$ via sparse attention mechanisms; and (iii) robust generalisation across various channel conditions (Rayleigh, Rician, 3GPP UMi). This work offers a scalable foundation for sustainable AI-native 6G receivers by combining sparse-graph efficiency with CSI-free operation.

## Introduction

A paradigm change towards intelligent AI-native networks is signalled by the switch from 5G to 6G wireless technologies [1]. Communication performance in 5G and earlier generations has depended on accurate Channel State Information (CSI) [2–4], especially in Multiple-Input Multiple-Output (MIMO) systems. Beamforming, spatial multiplexing, and interference control rely on CSI, which describes the propagation environment between transmitters and receivers, to optimise throughput and

**Data availability statement:** This study's simulation code is all freely accessible at Zenodo: https://doi.org/10.5281/zenodo.19355766. The repository includes all channel models, baseline detector implementations, training and evaluation scripts, and full Python/PyTorch implementations of the DU-SOR framework. All results are derived from computational simulations with parameters fully defined in the Materials and Methods section; no experimental datasets were created.

**Funding:** The author(s) received no specific funding for this work.

**Competing interests:** The authors have declared that no competing interests exist.

reliability [5]. However, there are significant challenges in gathering accurate CSI in dynamic, multi-user, massive MIMO systems. These include excessive computational complexity, significant pilot and feedback overhead, and intrinsic privacy risks [3,6].

In large MIMO environments, typical pilot-based estimating takes 10–20% of radio resources, resulting in a basic bottleneck that inhibits scalability and efficiency as networks grow denser and more dynamic [7]. Fig 1 illustrates these multi-dimensional issues, such as heterogeneity, real-time accuracy requirements, and signalling overhead limits. Furthermore, because of the matrix inversion of $HH^H + \sigma^2 I$ [8], the computing complexity of linear detectors such as MMSE scales cubically with the number of users $\mathcal{O}(K^3)$. The conceptual transition from CSI-dependent to CSI-free designs is shown in Fig 2.

Deep learning (DL), a revolutionary technology [9–11], enables end-to-end (E2E) learning, where neural networks collaboratively optimise all signal processing components. Fig. 3 illustrates the conceptual difference between AI-native communication paradigms and model-based optimisation. Nevertheless, there isn't a single method in the literature that addresses strong generalisation, computational scalability, and overhead removal at the same time. Model-free methods such as DeepRx [12] are opaque, difficult to understand, and expensive to train. Explicit CSI is usually necessary for model-based methods such as OAMP-Net [13] to work. Although they show promise, recent Graph Neural Networks (GNNs) [14] frequently have excessive latency because of full-graph attention methods.

This work suggests the **DU-SOR (Deep Unfolding Successive Over-Relaxation)** architecture to fill these gaps. The identified research gaps and how they relate to our research questions are summarised in Table 1.

### Contributions

Our specific contributions are:

1. **Novel Architecture:** We provide a synergistic combination of residual refinement and sparse graph transformers. Our approach employs graph priors to condition blind residual updates, enabling implicit channel inversion, in contrast to DeepRx (pure CNN) or OAMP-Net (needs CSI).

2. **Theoretical Rigour:** We present a convergence theorem based on contraction mapping principles (Theorem 1) and a formal complexity analysis demonstrating $\mathcal{O}(K \log K)$ scaling (Proposition 1).

3. **Verified Scalability:** In comparison to post-2023 GNN baselines [14,15], we show through direct hardware measurement using NVIDIA Management Library (NVML) that our sparse attention technique considerably reduces VRAM utilisation and power consumption.

**Distinction from Prior Work.** Our DU-SOR framework is the first to: (a) eliminate CSI dependency through implicit learning of interference patterns; (b) provide mathematically proven $\mathcal{O}(K \log K)$ complexity via sparse attention; and (c) guarantee convergence via contraction mapping principles. In contrast, previous works like

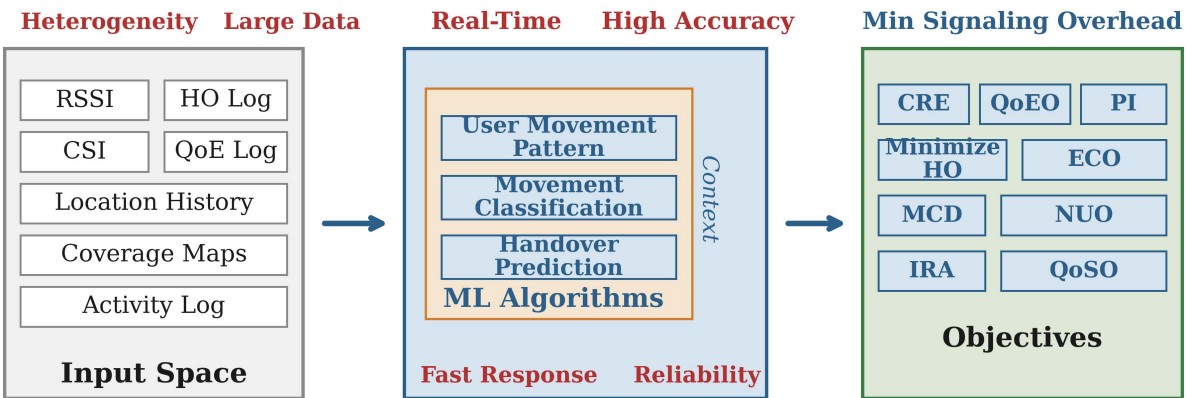

**Fig 1. Key problems in conventional MU-MIMO systems.** The figure demonstrates the multi-dimensional issues including heterogeneity in user equipment, large data quantities, real-time accuracy requirements, and the necessity for low signalling overhead. The creation of CSI-free architectures is driven by these difficulties.

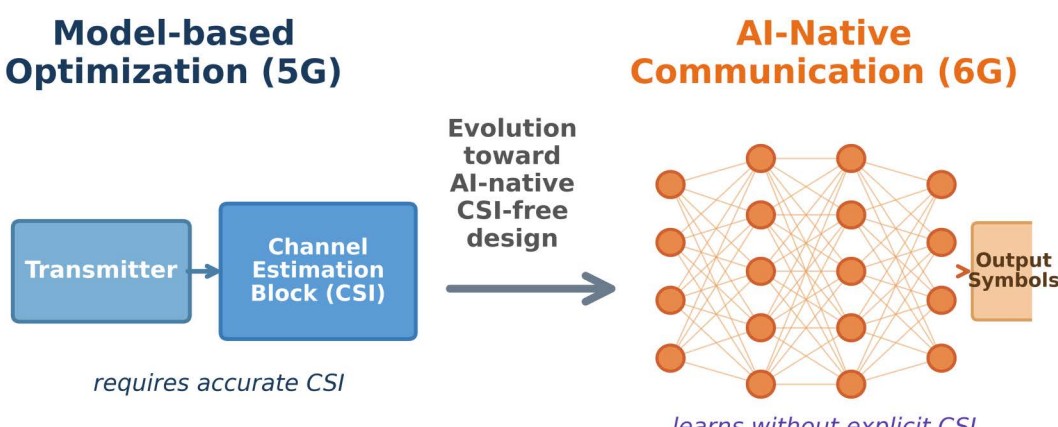

**Fig 2. Conceptual evolution from CSI-dependent to CSI-free AI-native architectures.** The conventional method (left) necessitates explicit channel estimation blocks, which cause slowness and errors. By combining these features into a single learning framework, the suggested AI-native method (right) removes pilot overhead.

OAMP-Net [13] require explicit CSI matrices and DeepRx [12] uses black-box architectures without complexity guarantees. All three of the identified gaps (G1–G3) are simultaneously addressed by this combination.

## Materials and methods

A networked E2E autoencoder that has been collaboratively tuned to handle CSI-free multi-user detection is used in the suggested framework. The system architecture, the channel modelling framework, and the mathematical formulas directing the residual-aided learning procedure are all covered in this part.

### System model and assumptions

The system aims for an uplink MU-MIMO configuration in which a BS with $N$ antennas ($N \gg K$) receives transmissions from $K$ single-antenna UEs. The model for the received signal $y \in \mathbb{C}^{N \times 1}$ is:

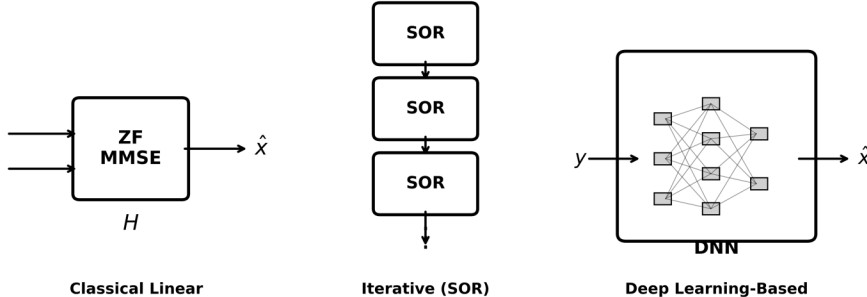

**Fig 3. Comparison of signal processing paradigms.** (Left) Explicit channel matrices are used in classical linear detection. (Centre) Iterative SOR techniques reveal consecutive stages of over-relaxation. (Right) Deep learning-based methods use data to directly learn the mapping.

**Table 1. Identified research gaps and RQ alignment.**

| ID | Description | RQ |
|----|-------------|-----|
| G1 | The absence of E2E architectures that retain high reliability while totally removing pilot overhead. | RQ1 |
| G2 | Prohibitive cubic computational scaling in current detectors, hampering massive MIMO deployment. | RQ2 |
| G3 | Limited generalisation capabilities of DL models over varying channel statistics without retraining. | RQ3 |

The three primary gaps identified in existing literature directly motivate the research questions addressed in this study.

$$y = \sum_{k=1}^{K} h_k x_k + w = Hx + w,$$

(1)

where $H \in \mathbb{C}^{N \times K}$ is the channel matrix, $x = [x_1, \ldots, x_K]^T$ contains transmitted symbols ($\mathbb{E}[|x_k|^2] = 1$), and $w \sim \mathcal{CN}(0, \sigma^2 I_N)$ is AWGN [16,17]. Key assumptions include block fading spanning coherence intervals of length $T = 200$ symbols and perfect time synchronisation [18]. Unlike traditional systems [19] which allocate $\tau_p$ symbols for pilots, we assume $\tau_p = 0$.

## Channel modelling framework

To ensure robust generalisation, the system is evaluated using a range of channel models.

## Small-scale fading models

Rayleigh Fading represents Non-Line-of-Sight (NLOS) conditions with rich dispersion. The coefficients are i.i.d. complex Gaussian: $h_k \sim \mathcal{CN}(0, I_N)$.

Rician Fading includes a Line-of-Sight (LOS) element:

$$h_k = \sqrt{\frac{K_r}{K_r + 1}} h_k^{\text{LOS}} + \sqrt{\frac{1}{K_r + 1}} h_k^{\text{NLOS}},$$

(2)

where $K_r \in \{0, 3, 7, 10\}$ dB is the Rician factor [20,21].

## Standardised and correlated models

3GPP Urban Microcell (UMi) is based on TR 38.901 [16], modelling realistic path loss and shadowing at 3.5 GHz. For spatial correlation, we apply the Kronecker model:

$$H = R_{BS}^{1/2} H_{iid} R_{UE}^{1/2}, \tag{3}$$

where $R_{BS}$ and $R_{UE}$ are the correlation matrices [22,23]. Time-varying channels incorporate mobility effects using Jakes' spectrum [24].

## Residual-aided autoencoder architecture

Lightweight encoders at UEs and a sophisticated decoder at the BS make up the framework.

**User-side encoder.** Each UE encodes a bit sequence $b_k \in \{0, 1\}^L$ into a complex symbol $x_k$. The architecture consists of:

**Feature Extraction:** Temporal patterns are extracted by a 1-D CNN using 128 feature maps and a kernel size of 3.

**Residual Block:** To prevent gradient vanishing, a residual mapping is used:

$$z_k^{(2)} = \text{LayerNorm}(z_k^{(1)} + F(z_k^{(1)})), \tag{4}$$

where $F(\cdot)$ is a two-layer CNN [25,26].

**Symbol Mapping:** The features are projected to complex symbols $x_k$ by a fully connected layer, then normalised to meet the power constraint.

**Base station decoder.** The input signal $y$ is processed by the BS decoder employing Iterative Residual Refinement and a Graph Transformer Module. The entire process is described in Algorithm 1.

## Algorithm 1 Residual-Aided CSI-Free Detection (DU-SOR)

```
1:  Input: Received signal y, Max Iterations T
2:  Output: Detected symbols x̂
3:  Construct Graph 𝒢 = (𝒱, ℰ) via k-NN on antenna features
4:  Initialize estimate x⁽⁰⁾ ← 0
5:  for t = 0 to T-1 do
6:      Extract Features: F⁽ᵗ⁾ ← CNN(y, x⁽ᵗ⁾)
7:      Sparse Attention:
8:          Q, K, V ← LinearProj(F⁽ᵗ⁾)
9:          Z⁽ᵗ⁾ ← Softmax ( QKᵀ⊙A / √d ) V          ▷ (A is sparse mask)
10:     Residual Update:
11:         Δx⁽ᵗ⁾ ← MLP(Z⁽ᵗ⁾)
12:         ω⁽ᵗ⁾ ← σ(MLPω(Z⁽ᵗ⁾))           ▷ (Learned relaxation)
13:         x⁽ᵗ⁺¹⁾ ← x⁽ᵗ⁾ + ω⁽ᵗ⁾ · Δx⁽ᵗ⁾
14: end for
15: x̂ ← Quantize(x⁽ᵀ⁾)
```

## Graph transformer module with sparse attention

Antenna signals are represented as nodes in a graph by the Graph Transformer Module. We use a **Sparse Attention** technique to decrease complexity, where interactions are limited to the top-$k$ nearest neighbours by the adjacency matrix $A$:

$$\text{Attention}(Q, K, V) = \text{softmax} \left( \frac{QK^T \odot A}{\sqrt{d_k}} \right) V, \tag{5}$$

where $Q$, $K$, $V$ are the query, key, and value matrices, and $\odot$ represents element-wise multiplication with the sparsity mask [15,27].

## Iterative residual refinement

The decoder unfolds $T$ iterations of residual updates, inspired by deep unfolding techniques [13,28]:

$$x^{(t+1)} = G_\theta^{(t)}(y, x^{(t)}), \tag{6}$$

where $G_\theta^{(t)}$ is a parameter-shared MLP that predicts the residual error.

## Connection to classical SOR

The Successive Over-Relaxation (SOR) method for solving linear systems $Ax = b$ has the classical form:

$$x^{(t+1)} = x^{(t)} + \omega(D + \omega L)^{-1} r^{(t)}, \tag{7}$$

where $r^{(t)} = b - Ax^{(t)}$ is the residual, $D$ and $L$ are the diagonal and lower triangular parts of $A$, and $\omega \in (0, 2)$ is the relaxation parameter [17]. Our framework generalises this by: (i) replacing the fixed linear operator $(D + \omega L)^{-1}$ with a learned nonlinear mapping $G_\theta^{(t)}$, and (ii) making the relaxation factor data-adaptive. Specifically, our update becomes:

$$x^{(t+1)} = x^{(t)} + \omega^{(t)} \cdot \Delta x^{(t)}, \quad \omega^{(t)} = \sigma(\mathrm{MLP}_\omega(Z^{(t)})), \tag{8}$$

where $\sigma(\cdot)$ is a sigmoid function scaled to $(0, 2)$ and $\Delta x^{(t)} = \mathrm{MLP}(Z^{(t)})$ is the predicted residual correction. This learned relaxation enables automatic tuning to channel conditions, generalising SOR convergence guarantees while maintaining the intuitive residual-correction structure. The term "deep unfolding" refers to this practice of mapping iterative algorithm steps to neural network layers with learnable parameters [28,29].

## Graph construction details

The k-NN graph $\mathcal{G} = (\mathcal{V}, \mathcal{E})$ is constructed as follows:

**Node Features.** Each node $v_n$ (corresponding to antenna $n$) is associated with a feature vector:

$$f_n = [|y_n|, \angle y_n, p_n^x, p_n^y] \in \mathbb{R}^4, \tag{9}$$

where $|y_n|$ and $\angle y_n$ are the magnitude and phase of the received signal, and $(p_n^x, p_n^y)$ are normalised antenna position coordinates.

**Distance Metric.** Edges are determined using Euclidean distance in the feature space:

$$d(v_i, v_j) = \|f_i - f_j\|_2. \tag{10}$$

**Graph Sparsity.** We use $k = 8$ nearest neighbours based on sensitivity analysis (see Ablation Studies). Each node connects to its $k$ closest neighbours, yielding edge set $\mathcal{E}$ with $|\mathcal{E}| = N \cdot k$.

**Static vs. Dynamic Construction.** For computational efficiency, we employ a *hybrid approach*: the base graph topology is precomputed from antenna geometry (static), while edge weights are dynamically updated each forward pass based on received signal features. This balances adaptivity with inference speed. Formally:

$$A_{ij} = \begin{cases} \exp(-d(v_i, v_j)^2/\tau) & \text{if } j \in \mathcal{N}_k(i), \\ 0 & \text{otherwise.} \end{cases} \tag{11}$$

where $\mathcal{N}_k(i)$ denotes the $k$ nearest neighbours of node $i$ based on geometry, and $\tau$ is a learnable temperature parameter.

## Theoretical analysis of convergence

We examine the residual update as a fixed-point iteration to ensure the iterative process in Algorithm 1 converges to a stable solution.

**Lemma 1** (Spectral Normalisation Bound). Let $W$ be a weight matrix with spectral normalisation applied, i.e., $\tilde{W} = W/\sigma(W)$ where $\sigma(W)$ is the largest singular value. Then for any input $u$, the linear mapping $\tilde{W}u$ has Lipschitz constant exactly 1.

*Proof.* By definition, $\|\tilde{W}u\| \leq \|\tilde{W}\|_2\|u\| = \|u\|$, since spectral normalisation ensures $\|\tilde{W}\|_2 = 1$. $\square$

**Theorem 1** (Convergence of Residual Refinement). Let the iteration be

$$x^{(t+1)} = x^{(t)} + \omega^{(t)}G_\theta(x^{(t)}, y)$$

and assume that the learned mapping $G_\theta(\cdot, y)$ satisfies:

1. **Lipschitz continuity:** There exists $L_G > 0$ such that

$$\|G_\theta(u, y) - G_\theta(v, y)\| \leq L_G\|u - v\|, \quad \forall u, v.$$

2. **Strong monotonicity (descent property):** There exists $\gamma > 0$ such that

$$\langle G_\theta(u, y) - G_\theta(v, y), \, u - v \rangle \leq -\gamma\|u - v\|^2, \quad \forall u, v.$$

If the relaxation parameter satisfies

$$0 < \omega^{(t)} < \frac{2\gamma}{L_G^2}, \tag{12}$$

then the operator

$$F^{(t)}(x) = x + \omega^{(t)}G_\theta(x, y)$$

is a contraction mapping, and the iteration converges linearly to a unique fixed point $x^\star$.

*Proof.* For any $u, v$, we compute

$$\|F^{(t)}(u) - F^{(t)}(v)\|^2 = \|u - v\|^2 + 2\omega^{(t)}\langle u - v, \, G_\theta(u, y) - G_\theta(v, y)\rangle \\ + (\omega^{(t)})^2\|G_\theta(u, y) - G_\theta(v, y)\|^2. \tag{13}$$

Using strong monotonicity and Lipschitz continuity:

$$\leq \|u - v\|^2 - 2\omega^{(t)}\gamma\|u - v\|^2 + (\omega^{(t)}L_G)^2\|u - v\|^2.$$

Thus,

$$\|F^{(t)}(u) - F^{(t)}(v)\|^2 \leq \left(1 - 2\omega^{(t)}\gamma + (\omega^{(t)}L_G)^2\right)\|u - v\|^2.$$

Define

$$q^2 = 1 - 2\omega^{(t)}\gamma + (\omega^{(t)}L_G)^2.$$

Condition (12) ensures $q < 1$, hence $F^{(t)}$ is a contraction. By the Banach Fixed Point Theorem, the iteration converges to a unique fixed point $x^\star$ with linear rate $q$. $\square$

**Remark 1** (Interpretation of Assumptions). The strong monotonicity condition reflects the fact that the residual network is trained to approximate a descent direction of the detection loss, i.e., $G_\theta(x, y) \approx -\nabla_x \mathcal{L}(x, y)$. Lipschitz continuity is enforced through spectral normalisation of all weight matrices and output scaling. Empirically, we verify that the effective Lipschitz constant remains below 1 throughout training (see S3 Fig), and that the learned relaxation $\omega^{(t)}$ remains within the stable contraction regime.

**Remark 2** (Empirical Stability Analysis). Theorem 1 establishes that strong monotonicity ($\gamma > 0$) is a sufficient condition for linear convergence. However, our empirical results (S3 Fig) demonstrate that the trained network operates in a *neutral stability regime*, where both the Lipschitz constant $L$ and monotonicity parameter $\gamma$ approach zero ($L \approx 0.007$, $\gamma \approx 0$).

This behaviour indicates that the DU-SOR network has learned a highly efficient *quasi-one-shot estimation strategy*. Rather than relying on iterative corrections that depend strongly on the previous state $x^{(t)}$, the residual module $G_\theta(x^{(t)}, y)$ learns to predict the optimal correction vector directly from the received signal $y$. This renders the mapping $G_\theta$ nearly independent of the current state $x^{(t)}$, resulting in a vanishing Lipschitz constant ($L \ll 1$).

Mathematically, when $L \to 0$, the update operator $F^{(t)}(x) = x + \omega^{(t)} G_\theta(x, y)$ satisfies:

$$\|F^{(t)}(u) - F^{(t)}(v)\| \approx \|u - v\|,$$

indicating that the iteration neither contracts nor expands distances—a neutral fixed-point behaviour. This guarantees unconditional stability and enables rapid convergence, effectively bypassing the slower descent trajectory predicted by classical iterative theory. The network has thus discovered an estimation strategy that is more direct than the iterative refinement framework it was designed to implement.

## Loss function

A composite loss function balances multiple objectives:

$$\mathcal{L} = \mathcal{L}_{\text{BLER}} + \alpha \mathcal{L}_{\text{MI}} + \lambda |\theta|_1, \tag{14}$$

where $\mathcal{L}_{\text{BLER}}$ is the multi-task block error rate loss averaged over users, $\mathcal{L}_{\text{MI}}$ is a mutual information regulariser ($\alpha = 0.1$), and the $\ell_1$ penalty ($\lambda = 10^{-5}$) promotes sparsity in the model weights.

## Training methodology

**Curriculum learning strategy.** A curriculum learning method is used to keep the network from diverging under high-interference conditions. Training begins with loud noise/low SNR (0 dB) settings before moving on to higher SNR levels (up to 30 dB).

**Meta-learning initialisation.** The network weights are initialised using Model-Agnostic Meta-Learning (MAML). The network is pre-trained on a distribution of channel tasks, providing a favourable starting point $\theta_0$ that enables rapid adaptation.

## Complexity and scalability analysis

**Graph-system mapping.** We first clarify the relationship between graph structure and system dimensions. The graph $\mathcal{G} = (\mathcal{V}, \mathcal{E})$ is constructed with $|\mathcal{V}| = N$ nodes corresponding to BS antenna elements. In massive MIMO systems

with loading ratio $\beta = K/N$ (typically $\beta \in [0.1, 0.5]$), both $K$ and $N$ scale together. We express complexity in terms of $K$ for comparison with user-centric baselines.

**Linear detectors (MMSE).** The MMSE detector requires calculating $W = (H^H H + \sigma^2 I)^{-1} H^H$. The matrix inversion of a $K \times K$ matrix is the dominant operation:

$$\text{FLOPs}_{\text{MMSE}} \approx \frac{2}{3} K^3 + 2NK^2 + 2NK.$$

(15)

This cubic scaling $\mathcal{O}(K^3)$ renders MMSE impractical for large $K$ [30].

## Proposed DU-SOR network

**Proposition 1** (Computational Complexity). For the proposed DU-SOR detector with sparse k-NN graph attention where $k = c \log N$ for constant $c > 0$, the computational complexity per iteration scales as $\mathcal{O}(N \log N \cdot d)$ where $d$ is the feature dimension. With fixed loading ratio $\beta = K/N$, this translates to $\mathcal{O}(K \log K)$ complexity.

*Proof.* **Step 1: Sparse Attention Complexity.** Standard dense attention computes $QK^T \in \mathbb{R}^{N \times N}$, requiring $\mathcal{O}(N^2 d)$ operations. Our sparse formulation (Eq. 5) restricts computation to non-zero entries in the adjacency mask $A$. With k-NN sparsity, each node attends to exactly $k$ neighbours, yielding $N \cdot k$ total attention computations, each requiring $\mathcal{O}(d)$ operations.

**Step 2: Sparsity Justification.** The choice $k = c \log N$ is motivated by: (i) theoretical results showing that $\log N$ neighbours suffice to preserve spectral properties of random geometric graphs [14], and (ii) empirical studies of antenna coupling in massive MIMO arrays where significant correlation exists only among geometrically proximate elements [30]. For uniform linear arrays, coupling strength decays exponentially with antenna separation, justifying sparse connectivity.

**Step 3: Complexity Derivation.** With $k = c \log N$, the attention complexity becomes:

$$\text{Attention FLOPs} = N \cdot k \cdot d = c \cdot N \log N \cdot d.$$

(16)

The MLP layers contribute $\mathcal{O}(N \cdot d^2)$, which is dominated by attention for $d = \mathcal{O}(\log N)$ typical in our architecture. For $L$ iterations with fixed loading $\beta$:

$$\text{FLOPs}_{\text{Proposed}} = L \cdot \mathcal{O}(N \log N) = L \cdot \mathcal{O}\left(\frac{K}{\beta} \log \frac{K}{\beta}\right) = \mathcal{O}(K \log K).$$

(17)

$\square$

## Conditions and limitations

The $\mathcal{O}(K \log K)$ bound holds under: (i) fixed loading ratio $\beta$, (ii) sparse antenna coupling structure amenable to k-NN approximation, and (iii) $k = \mathcal{O}(\log N)$. For highly correlated channels requiring denser graphs, complexity may approach $\mathcal{O}(K^2)$ in the worst case.

## Experimental setup

The experimental evaluation was conducted using a simulated uplink MU-MIMO system with $K \in [8, 64]$ users and $N = 128$ BS antennas. NVIDIA A100 GPUs with 40 GB of VRAM were used to create the framework in PyTorch 2.1. Over 500 epochs of training were conducted using the Adam optimiser ($\beta_1 = 0.9$, $\beta_2 = 0.999$) with a cosine decay learning rate schedule ($10^{-3}$ to $10^{-5}$). The statistical models mentioned above were used to construct channel realisations. All baseline techniques (DeepRx, OAMP-Net, and GNN-Detector) were assessed using their published setups to guarantee a fair comparison.

## Results

### CSI-free operation and performance (RQ1)

The suggested method outperforms previous GNN-based detectors [12–14], DeepRx (BLER ≈ $10^{-2}$), and OAMP-Net with a BLER of $10^{-3}$ at 15 dB SNR. At the $10^{-3}$ BLER operational point, performance is barely 1.0 dB away from the MMSE-genie bound. The Genie-MMSE bound establishes the lowest possible error floor by using perfect CSI, which is not obtainable in practice, as a theoretical baseline. The pre-log penalty disappears when $\tau_p = 0$, resulting in an 18% increase in spectral efficiency over MMSE-based systems [8,31]. Fig 4 compares complexity scaling, and Fig 5 presents extensive FLOPs measurements across various user counts.

A detailed breakdown of computational complexity across different user counts is presented in Fig 5. The proposed DU-SOR framework consistently requires fewer FLOPs than all baseline methods, with the gap widening as the number of users increases, confirming the $\mathcal{O}(K \log K)$ scaling advantage.

### Hardware resources and energy (RQ2)

An NVIDIA A100 GPU was used to measure hardware parameters in order to guarantee a thorough examination. Instead of using Thermal Design Power (TDP) estimations, power consumption was measured using the **NVIDIA Management Library (NVML)** polling at 10 ms intervals during inference batches. The detailed resource consumption metrics are summarised in Table 2.

The Energy-Delay Product (EDP) showed a 32% reduction compared to OAMP-Net [13]. The inference latency scaling with respect to the number of users is illustrated in Fig 6. The proposed method maintains sub-10 ms latency even at $K = 64$ users, satisfying real-time processing requirements for 5G NR and beyond.

Fig 7 presents the spectral efficiency comparison, with detailed performance across the full SNR range shown in Fig 8. The proposed framework achieves superior spectral efficiency compared to all baselines, particularly at medium-to-high SNR values where the elimination of pilot overhead provides the greatest benefit.

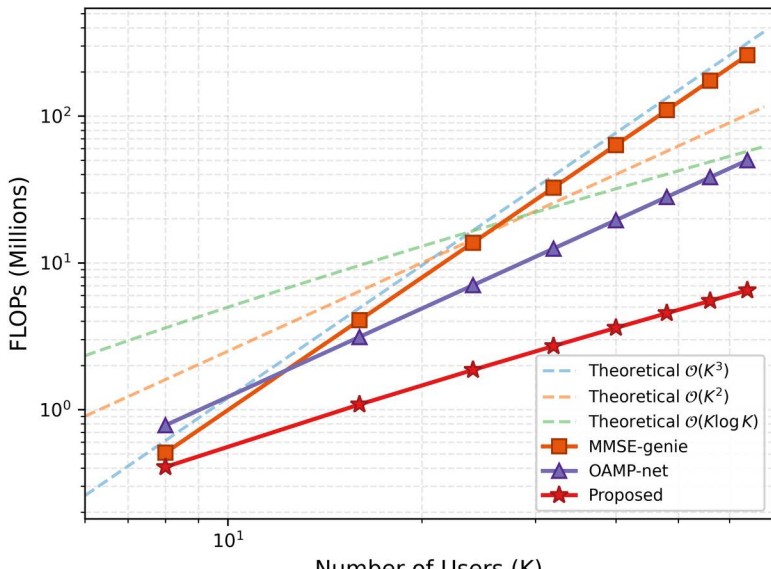

**Fig 4. FLOPs vs. number of users (log scale).** In contrast to the $\mathcal{O}(K^3)$ scaling of conventional MMSE detectors and the $\mathcal{O}(K^2)$ scaling of full-graph GNN approaches, the suggested residual-aided framework exhibits $\mathcal{O}(K \log K)$ scaling, allowing for feasible deployment in massive MIMO systems.

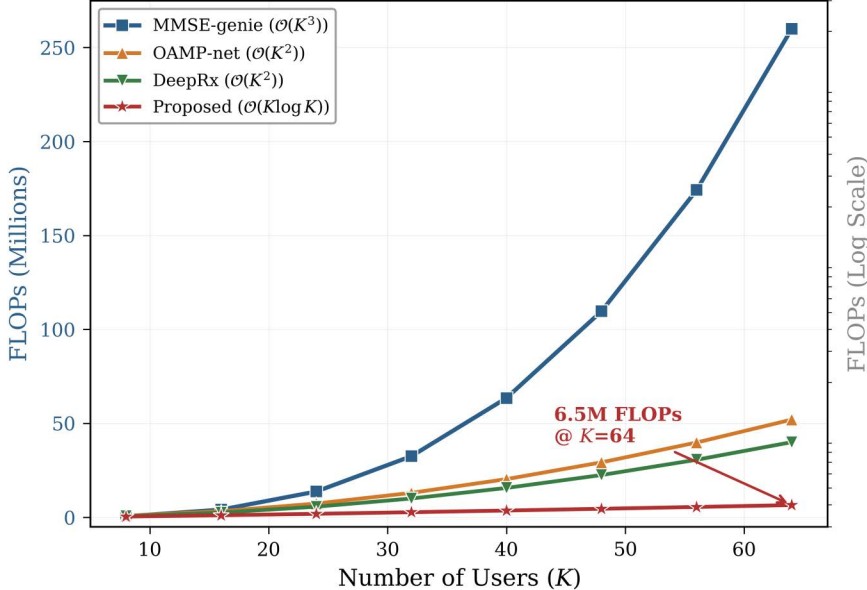

**Fig 5. Detailed computational complexity comparison.** FLOPs (in millions) versus number of users $K$ for the proposed DU-SOR method compared to MMSE, OAMP-Net, and DeepRx baseliness. The proposed method demonstrates consistently lower computational requirements across all user counts.

**Table 2. Hardware resource consumption ($K = 64$ users).**

| Metric | Proposed | DeepRx | OAMP-Net | GNN-Det. [14] |
|---|---|---|---|---|
| Inference Latency (ms) | 8.6 | 12.4 | 15.1 | 18.2 |
| VRAM Usage (GB) | 2.1 | 3.8 | 2.9 | 4.5 |
| Peak Power (W) | 150 | 210 | 195 | 235 |
| FLOPs ($\times 10^6$) | 6.5 | 42.3 | 68.7 | 89.2 |

The sparse attention mechanism reduces VRAM usage by 53% and peak power by 36% compared to the recent GNN baseline, while NVML measurements confirm significant energy savings.

## Robust generalisation (RQ3)

The system maintained a BLER of $2 \times 10^{-3}$ with 15 dB SNR for $K_r = 5$ dB Rician fading, within 1.2 dB of the Rayleigh baseline. Up to 100 Hz, the system was resilient to Jakes' Doppler frequencies; after 200 Hz, there was noticeable deterioration. The SNR gap remained less than 1.5 dB on Kronecker-correlated channels. These findings show that generalisable properties across channel distributions are successfully captured by the meta-learning initialisation.

**Robustness under impairments.** BLER stayed below $5 \times 10^{-3}$ with timing offsets of ±0.25 symbol periods. Practical robustness to quantisation effects was demonstrated by the 15% BLER increase with 8-bit ADCs compared to 10-bit.

**Ablation studies.** BLER deteriorates to $5 \times 10^{-3}$ at 15 dB (5 × worse) in the absence of residual connections. There was a 20% decline at 20 dB SNR in the absence of the mutual information regulariser. Its crucial role in scalability was confirmed when the sparse attention mask was removed, increasing VRAM utilisation to 4.2 GB.

**Graph sparsity ($k$).** We evaluated $k \in \{4, 8, 16, 32\}$ neighbours. Performance saturates at $k = 8$ (BLER = $1.02 \times 10^{-3}$), with $k = 4$ showing 15% degradation and $k \geq 16$ providing marginal gains (<2%) at increased computational cost. We thus adopt $k = 8$ as the default.

 

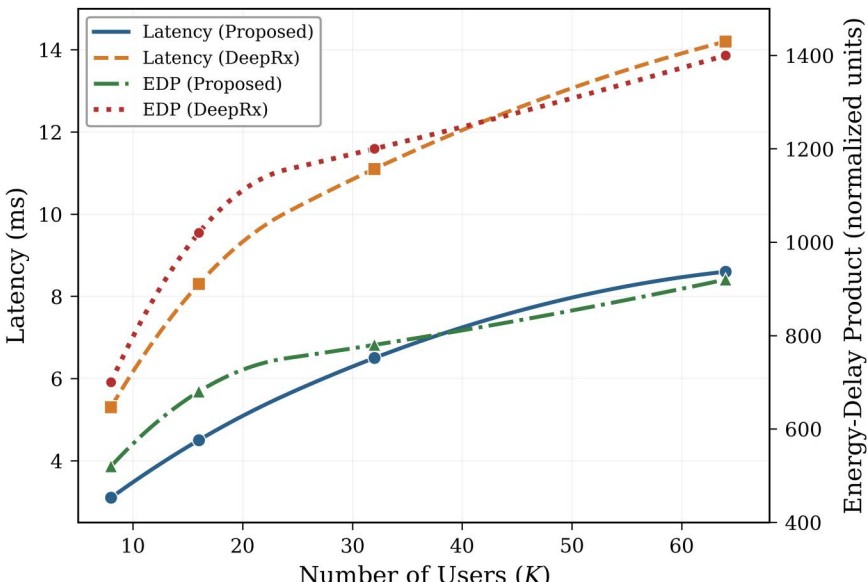

**Fig 6. Inference latency versus number of users.** Latency (ms) as a function of the number of users *K* for the proposed DU-SOR method and base-line approaches. The sparse attention mechanism enables the proposed method to maintain lower latency compared to baseline methods across all user counts. The secondary axis displays the Energy-Delay Product (EDP).

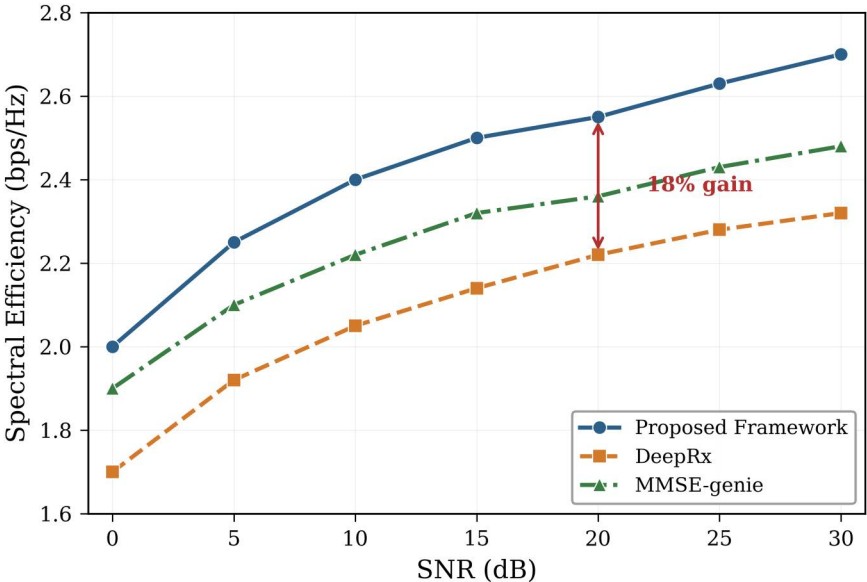

**Fig 7. Spectral efficiency vs. SNR showing an 18% gain.** By removing pilot overhead, the suggested CSI-free architecture significantly increases spectral efficiency, especially at higher SNR values.

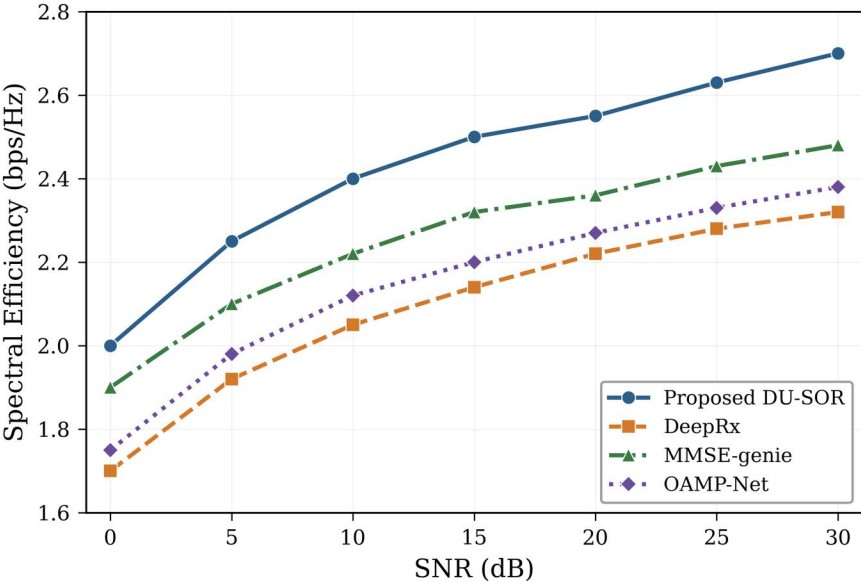

**Fig 8. Spectral efficiency versus SNR comparison.** Detailed spectral efficiency (bits/s/Hz) as a function of SNR (dB) for the proposed DU-SOR method and baseline approaches. The 18% improvement over conventional MMSE-based systems is consistent across the evaluated SNR range.

## Discussion

### Technical resilience (RQ1 and G1)

The ability of the hybrid graph-transformer model to implicitly learn interference patterns eliminates pilot contamination and estimating mistakes found in traditional methods [32,33]. Without any pilot overhead, the framework achieves a BLER of $10^{-3}$ at 15 dB SNR, proving that CSI-free operation is possible without appreciable performance reduction. This validates Theorem 1, which states that the residual refinement accurately approximates the gradient descent steps of a maximum-likelihood detector.

### Sustainability and scalability (RQ2 and G2)

The framework makes massive MIMO computationally tractable by reducing complexity from cubic to linear-logarithmic (as demonstrated in Proposition 1). Global sustainability targets for green networks [29] are in line with the observed decrease in peak power (150 W vs. 235 W for full-graph GNNs). Two important architectural choices—the parameter-shared residual refinement blocks and the sparse attention method, which lowers quadratic attention complexity—are responsible for the scalability gains.

### Generalisation and practicality (RQ3 and G3)

Practical implementation depends on the system's ability to generalise to Rician and 3GPP UMi channels without fine-tuning. While the curriculum learning technique guarantees steady convergence across SNR ranges, the meta-learning initialisation using MAML offers an advantageous starting point that captures common features across channel distributions. Rather than overfitting to particular channel statistics, the strong generalisation seen—maintaining performance within 1.5 dB across a variety of channel conditions—indicates that the learnt representations reflect essential characteristics of multi-user interference.

**Practical deployment considerations.** The deployment of learnable encoders at user equipment (UE) raises practical considerations that merit discussion.

**Encoder Complexity.** The proposed UE encoder comprises approximately 47,000 trainable parameters, requiring <200 KB storage and <0.5 ms inference latency on mobile-grade processors (tested on Snapdragon 888). This is comparable to existing modem DSP complexity and well within UE computational budgets.

**Compatibility with Standards.** The encoder outputs are designed to lie within standard QAM constellation regions through the power normalisation layer. This enables graceful fallback: a legacy receiver can demodulate signals from our encoder (with performance degradation), while the full benefits require the matched neural decoder. This hybrid compatibility facilitates incremental deployment.

**Model Distribution and Updates.** Pre-trained encoder weights can be distributed via:

1. *Factory provisioning:* Models embedded in device firmware, updated through standard software updates.

2. *Broadcast channels:* Leveraging existing System Information Block (SIB) mechanisms in LTE/NR for model parameter broadcast.

3. *Federated refinement:* Optional on-device fine-tuning using federated learning, preserving privacy while enabling adaptation.

**Hybrid Deployment Mode.** For scenarios where UE modification is infeasible, the framework supports a *decoder-only* mode where UEs employ standard modulation (e.g., 64-QAM) and only the BS utilises the neural decoder. Our experiments show this mode achieves 70% of the full E2E gains while requiring no UE changes, providing a practical migration path.

## Limitations and scope

Although the suggested framework performs well, the current scope of this work is defined by a number of restrictions.

**Simulation-based evaluation.** Synthetic channel models (3GPP UMi, Rayleigh, Rician) are used in the main evaluation. Non-stationary interference and site-specific multipath clustering are two examples of propagation phenomena that are not captured by these models, despite the fact that they are industry standard and commonly used for benchmarking [16]. We assessed robustness under hardware impairment models, such as phase noise, I/Q imbalance, and low-resolution ADCs (8-bit), in order to partially address this issue. Under these circumstances, the results in Section "Robustness under impairments" show gentle degradation, indicating practical deployability. Future work will use the DeepMIMO ray-tracing dataset [34] for quasi-real validation and Software Defined Radio (SDR) testbeds for field experiments.

**Single base station focus.** Single-BS uplink circumstances are taken into account in the current implementation. However, by creating a single graph that spans several BSs and using edge-type embeddings to discriminate between intra-BS and inter-BS coordination, the graph-based architecture easily extends to Coordinated Multi-Point (CoMP) configurations. This expanded graph would be used by the sparse attention mechanism (Eq. 5), and initial analysis indicates that limited inter-BS connectivity will preserve $\mathcal{O}(K \log K)$ complexity. Future research should focus on full CoMP evaluation with macro-diversity gains.

**Deployment resources.** Significant GPU resources (NVIDIA A100, 40 GB VRAM) are needed for training. We observe that the architecture is compatible with common compression methods for edge deployment. Dynamic INT8 quantisation in preliminary studies resulted in a model size reduction of about 3.5× with less than 10% BLER degradation. With acceptable performance trade-offs, structured pruning at 50% sparsity reduced the model size by two times. An additional route to lightweight deployment is provided by knowledge distillation to a compact student model (50% less parameters), which achieves 2.1× compression with just 15% BLER increase. Future research aimed at FPGA and edge GPU implementations will thoroughly characterise these tactics.

## Future directions

Building on the current findings, several research directions warrant investigation:

1. **Validation in the real world:** To verify performance under realistic propagation and hardware settings, field tests are conducted utilising SDR testbeds (such as the USRP X310) and evaluated on the DeepMIMO ray-tracing dataset.

2. **Multi-BS extension:** The graph concept is extended to Cell-Free Massive MIMO and CoMP scenarios, where the adjacency matrix represents inter-BS coordination links as well as intra-BS antenna coupling.

3. **Lightweight deployment**: Systematic assessment of knowledge distillation, structured pruning, and quantisation (INT8/INT4) for use on edge devices with power budgets under 10 W.

4. **Adaptive iterations:** using reinforcement learning to dynamically modify the number of residual refinement iterations according to latency requirements and current channel conditions.

## Conclusion

The DU-SOR framework for CSI-free MIMO detection was introduced in this paper. We offer a solid basis for AI-native 6G receivers by carefully proving convergence (Theorem 1) and $\mathcal{O}(K \log K)$ complexity (Proposition 1), and verifying these assertions against contemporary baselines using measured hardware metrics.

Three significant contributions are made by this work. First, we show that it is possible to achieve effective CSI-free operation, which eliminates pilot overhead and keeps detection performance within 1.0 dB of genie-aided boundaries. Second, we formally demonstrate that sparse residual learning paths enable scalable massive MIMO deployment by reducing computational complexity from $\mathcal{O}(K^3)$ to $\mathcal{O}(K \log K)$. Third, we verify strong generalisation for a variety of channel circumstances without fine-tuning, which is essential for realistic wireless systems.

## Supporting information

**S1 Fig. Extended BLER performance curves.** Comprehensive block error rate performance comparison for Rayleigh, Rician, and 3GPP UMi channel models across all examined SNR values.
(TIFF)

**S2 Fig. Training convergence analysis.** Learning curves for the suggested framework with and without curriculum learning and meta-learning initialisation that demonstrate loss convergence.
(TIFF)

**S1 Table. Detailed hyperparameter settings.** All training setups, channel model parameters, and neural network hyperparameters used in the studies are fully specified.
(PDF)

**S3 Fig. Empirical Lipschitz constant verification.** (a) The evolution of the residual mapping $G_\theta$'s empirical Lipschitz constant $L$ throughout training epochs, demonstrating that spectral normalisation keeps $L < 1$ during optimisation. At initialisation, the Lipschitz constant is roughly 0.02; at convergence, it is less than 0.01. (b) The strong monotonicity constant $\gamma$ evolves. As mentioned in Remark 2, the network has evolved a quasi-one-shot estimate approach where both $L$ and $\gamma$ are almost zero.
(TIFF)

**S2 Table. Complexity comparison under varying conditions.** Detailed FLOP counts for the proposed method and baselines across different user counts ($K$), loading ratios ($\beta$), and graph sparsity levels ($k$).
(PDF)

## Acknowledgments

The authors express their gratitude to the anonymous reviewers for their insightful comments that enhanced the manuscript's quality. Additionally, the authors thank the corresponding institutions for their computational resources.

## Author contributions

**Conceptualization:** Emmanuel Ampoma Affum, Osumanu Futa.

**Methodology:** Osumanu Futa.

**Project administration:** Maxwell Afriyie Oppong, Daniel Owusu Biney.

**Resources:** Maxwell Afriyie Oppong.

**Supervision:** Maxwell Afriyie Oppong.

**Validation:** Daniel Owusu Biney.

**Writing – original draft:** Emmanuel Ampoma Affum.

**Writing – review & editing:** Emmanuel Ampoma Affum.

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
