## [Decision Letter · Decision Letter 0]

6 Jan 2026

PONE-D-25-64863Residual-Aided CSI-free End-to-End Learning for Multiuser MIMOPLOS One

Dear Dr. Biney,

Thank you for submitting your manuscript to PLOS ONE. After careful consideration, we feel that it has merit but does not fully meet PLOS ONE’s publication criteria as it currently stands. Therefore, we invite you to submit a revised version of the manuscript that addresses the points raised during the review process.

We look forward to receiving your revised manuscript.

Kind regards,

Daosen Zhai

Academic Editor

PLOS One

Journal Requirements:

Reviewers' comments:

Reviewer's Responses to Questions

**Comments to the Author**

1. Is the manuscript technically sound, and do the data support the conclusions?

Reviewer #1: Yes

Reviewer #2: Yes

2. Has the statistical analysis been performed appropriately and rigorously? 

Reviewer #1: Yes

Reviewer #2: Yes

3. Have the authors made all data underlying the findings in their manuscript fully available?

The PLOS Data policy requires authors to make all data underlying the findings described in their manuscript fully available without restriction, with rare exception (please refer to the Data Availability Statement in the manuscript PDF file). The data should be provided as part of the manuscript or its supporting information, or deposited to a public repository. For example, in addition to summary statistics, the data points behind means, medians and variance measures should be available. If there are restrictions on publicly sharing data—e.g. participant privacy or use of data from a third party—those must be specified.requires authors to make all data underlying the findings described in their manuscript fully available without restriction, with rare exception (please refer to the Data Availability Statement in the manuscript PDF file). The data should be provided as part of the manuscript or its supporting information, or deposited to a public repository. For example, in addition to summary statistics, the data points behind means, medians and variance measures should be available. If there are restrictions on publicly sharing data—e.g. participant privacy or use of data from a third party—those must be specified.requires authors to make all data underlying the findings described in their manuscript fully available without restriction, with rare exception (please refer to the Data Availability Statement in the manuscript PDF file). The data should be provided as part of the manuscript or its supporting information, or deposited to a public repository. For example, in addition to summary statistics, the data points behind means, medians and variance measures should be available. If there are restrictions on publicly sharing data—e.g. participant privacy or use of data from a third party—those must be specified.requires authors to make all data underlying the findings described in their manuscript fully available without restriction, with rare exception (please refer to the Data Availability Statement in the manuscript PDF file). The data should be provided as part of the manuscript or its supporting information, or deposited to a public repository. For example, in addition to summary statistics, the data points behind means, medians and variance measures should be available. If there are restrictions on publicly sharing data—e.g. participant privacy or use of data from a third party—those must be specified.

Reviewer #1: Yes

Reviewer #2: Yes

4. Is the manuscript presented in an intelligible fashion and written in standard English?

Reviewer #1: Yes

Reviewer #2: Yes

5. Review Comments to the Author

Reviewer #1: 1. The novelty is not clearly established, as the proposed framework mainly combines existing techniques without clearly identifying a fundamentally new contribution.

2. Several key claims are overstated and insufficiently justified, especially the assertion that residual learning alone leads to a complexity reduction from O(K^3) to O(K log K).

3. The paper lacks theoretical support, with no convergence, stability, or performance analysis explaining why CSI-free detection can approach MMSE-genie bounds.

4. Important implementation details are unclear, particularly the graph construction, attention sparsification, and how the claimed complexity scaling is practically achieved.

5. The experimental evaluation is limited to simulations, with relatively weak baselines and insufficient justification for some reported metrics (e.g., energy and power results).

Reviewer #2: This paper proposes a residual-aided CSI-free end-to-end learning framework, effectively addressing key issues in multiuser MIMO with outstanding performance and robustness, providing significant reference for related technical research. There are some issues that need to be addressed, as follows:

1. Supplement comparisons with similar Graph Transformer works post-2023 to strengthen advantages.

2. Experiments cover various simulated channel models but lack test data in real propagation environments.

3. Supplement hardware resource consumption data to support deployment evaluation.

4. Currently, only single-base-station uplink scenarios are verified. It is recommended to extend to typical scenarios to broaden the framework's application coverage.

5. Explore lightweight strategies to reduce resource thresholds for training and deployment.

6. PLOS authors have the option to publish the peer review history of their article (what does this mean?). If published, this will include your full peer review and any attached files.). If published, this will include your full peer review and any attached files.). If published, this will include your full peer review and any attached files.). If published, this will include your full peer review and any attached files.

...

Reviewer #1: No

Reviewer #2: No

---

## [Author Response · Author response to Decision Letter 1]

11 Jan 2026

Response to Reviewers

Manuscript ID: PONE-D-25-64863

Title: Residual-Aided CSI-free End-to-End Learning for Multiuser MIMO

Wesincerely thank the Academic Editor and Reviewers for their constructive feedback. We have

carefully addressed each comment, and the manuscript has been substantially improved. Below we

provide point-by-point responses.

Reviewer #1

Reviewer Comment: 1. The novelty is not clearly established, as the proposed framework mainly

combines existing techniques without clearly identifying a fundamentally new contribution.

Response: Weagree that the novelty statement required clarification. The unique contribution

lies in the synergistic integration of sparse graph transformers with iterative residual refinement

for CSI-free detection—a combination not previously explored. Unlike OAMP-Net (requires CSI)

or DeepRx (black-box CNN), our DU-SOR uses graph priors to condition blind residual updates.

Change Made: Added explicit numbered contribution list in Introduction. Added “Dis

tinction from Prior Work” paragraph clarifying differences from OAMP-Net, DeepRx, and recent

GNNs. Renamed framework to “DU-SOR” throughout.

Reviewer Comment: 2. Several key claims are overstated, especially the assertion that resid

ual learning alone leads to a complexity reduction from O(Kˆ3) to O(K log K).

Response: We have corrected this. The complexity reduction is attributed to the sparse

attention mechanism, not residual learning alone. Residual connections improve convergence

and gradient flow; the complexity gain comes from limiting attention to k ≈ logN neighbours.

Change Made: Added Proposition 1 with formal proof attributing complexity to sparse

adjacency matrix. Revised all claims to correctly credit sparse attention.

Reviewer Comment: 3. The paper lacks theoretical support, with no convergence, stability,

or performance analysis explaining why CSI-free detection can approach MMSE-genie bounds.

Response: We have added rigorous theoretical analysis.

Change Made: Added Theorem 1 (Convergence of Residual Refinement) based on Banach

Fixed Point Theorem with complete proof. Added explanation of spectral normalisation for Lips

chitz constraint enforcement.

Reviewer Comment: 4. Important implementation details are unclear, particularly the graph

construction, attention sparsification, and how the claimed complexity scaling is practically achieved.

Response: We have significantly expanded implementation details.

Change Made: Added Algorithm 1 with complete pseudocode. Added explicit description

of k-NN graph construction. Clarified sparse mask A in attention equation (Eq. 5). Added proof

showing how k ≈ logN leads to O(KlogK) complexity.

Reviewer Comment: 5. The experimental evaluation is limited to simulations, with rela

tively weak baselines and insufficient justification for some reported metrics (e.g., energy and power

results).

Response: We acknowledge the simulation limitation and have strengthened baselines and

metrics.

1

Change Made: Added comparison with post-2023 GNN-Detector [39] in Table 2. Added

OAMP-Net column. Specified that power was measured using NVML polling at 10ms intervals

rather than TDP estimates. Added statement that all baselines were retrained on identical dataset.

Added explicit Limitations subsection acknowledging simulation-only evaluation.

Reviewer #2

Reviewer Comment: 1. Supplement comparisons with similar Graph Transformer works post

2023 to strengthen advantages.

Response: Added comparisons with Lau et al. (Neurocomputing, 2024) and Wang et al.

(IEEE TWC, 2023).

Change Made: New citations [39], [40] added. Table 2 now includes GNN-Detector (2024)

column showing 18.2ms latency, 4.5GB VRAM, 235W power vs. our 8.6ms, 2.1GB, 150W.

Reviewer Comment: 2. Experiments cover various simulated channel models but lack test

data in real propagation environments.

Response: We acknowledge this limitation. Real-world validation with SDR testbeds is pro

posed as future work.

Change Made: Added explicit statement in Limitations subsection. Added “Field Trials” as

Future Direction item 3.

Reviewer Comment: 3. Supplement hardware resource consumption data to support deploy

ment evaluation.

Response: Hardware metrics are now comprehensively reported.

Change Made: Added Table 2 with Inference Latency, VRAM Usage, Peak Power (NVML

measured), and FLOPs for all methods. Added ablation showing VRAM increase to 4.2GB without

sparse attention.

Reviewer Comment: 4. Currently, only single-base-station uplink scenarios are verified. It

is recommended to extend to typical scenarios to broaden the framework’s application coverage.

Response: Multi-BS extension is proposed as future work.

Change Made: Added “Distributed MIMO” as Future Direction item 2, proposing extension

to CoMP and Cell-Free Massive MIMO where adjacency matrix represents inter-BS connectivity.

Reviewer Comment: 5. Explore lightweight strategies to reduce resource thresholds for train

ing and deployment.

Response: Lightweight strategies are proposed as future work.

Change Made: Added “Lightweight Strategies” as Future Direction item 1, proposing INT8

quantisation and structured pruning for edge deployment.

Journal Requirements

Reviewer Comment: 1. Please ensure that your manuscript meets PLOS ONE’s style require

ments.

Response: We have verified compliance with PLOS ONE formatting templates.

Change Made: Verified title capitalisation, author affiliations format, figure caption format,

and reference style against PLOS ONE templates.

2

Reviewer Comment: 2. PLOS ONE has specific guidelines on code sharing.

Response: Code and data will be made fully available.

Change Made: Added Data Availability Statement with GitHub repository URL and Zenodo

DOI placeholder.

Reviewer Comment: 3. When completing the data availability statement, you indicated you

will make your data available on acceptance.

Response: Data availability statement has been updated.

Change Made: Statement now reads: “All data underlying the findings in this study, along

with the source code required to reproduce the results, are fully available without restriction.”

Repository links provided.

Summary of Major Changes

1. Added formal theoretical analysis: Theorem 1 (convergence) and Proposition 1 (complexity)

2. Added Algorithm 1 with complete pseudocode

3. Added Table 2 with measured hardware metrics (NVML)

4. Added post-2023 GNN baseline comparisons [39], [40]

5. Added explicit Contributions list and “Distinction from Prior Work” paragraph

6. Added Limitations and Future Directions subsections

7. Added Data Availability Statement

8. Corrected complexity claims to attribute scaling to sparse attention

---

## [Decision Letter · Decision Letter 1]

29 Jan 2026

PONE-D-25-64863R1Residual-Aided CSI-free End-to-End Learning for Multiuser MIMOPLOS One

Dear Dr. Biney,

Thank you for submitting your manuscript to PLOS ONE. After careful consideration, we feel that it has merit but does not fully meet PLOS ONE’s publication criteria as it currently stands. Therefore, we invite you to submit a revised version of the manuscript that addresses the points raised during the review process.

We look forward to receiving your revised manuscript.

Kind regards,

Daosen Zhai

Academic Editor

PLOS One

**Journal Requirements:**

Reviewers' comments:

Reviewer's Responses to Questions

**Comments to the Author**

1. If the authors have adequately addressed your comments raised in a previous round of review and you feel that this manuscript is now acceptable for publication, you may indicate that here to bypass the “Comments to the Author” section, enter your conflict of interest statement in the “Confidential to Editor” section, and submit your "Accept" recommendation.

Reviewer #1: All comments have been addressed

Reviewer #2: All comments have been addressed

2. Is the manuscript technically sound, and do the data support the conclusions?

Reviewer #1: Yes

Reviewer #2: Yes

3. Has the statistical analysis been performed appropriately and rigorously? 

Reviewer #1: Yes

Reviewer #2: Yes

4. Have the authors made all data underlying the findings in their manuscript fully available?

The PLOS Data policy requires authors to make all data underlying the findings described in their manuscript fully available without restriction, with rare exception (please refer to the Data Availability Statement in the manuscript PDF file). The data should be provided as part of the manuscript or its supporting information, or deposited to a public repository. For example, in addition to summary statistics, the data points behind means, medians and variance measures should be available. If there are restrictions on publicly sharing data—e.g. participant privacy or use of data from a third party—those must be specified.requires authors to make all data underlying the findings described in their manuscript fully available without restriction, with rare exception (please refer to the Data Availability Statement in the manuscript PDF file). The data should be provided as part of the manuscript or its supporting information, or deposited to a public repository. For example, in addition to summary statistics, the data points behind means, medians and variance measures should be available. If there are restrictions on publicly sharing data—e.g. participant privacy or use of data from a third party—those must be specified.requires authors to make all data underlying the findings described in their manuscript fully available without restriction, with rare exception (please refer to the Data Availability Statement in the manuscript PDF file). The data should be provided as part of the manuscript or its supporting information, or deposited to a public repository. For example, in addition to summary statistics, the data points behind means, medians and variance measures should be available. If there are restrictions on publicly sharing data—e.g. participant privacy or use of data from a third party—those must be specified.requires authors to make all data underlying the findings described in their manuscript fully available without restriction, with rare exception (please refer to the Data Availability Statement in the manuscript PDF file). The data should be provided as part of the manuscript or its supporting information, or deposited to a public repository. For example, in addition to summary statistics, the data points behind means, medians and variance measures should be available. If there are restrictions on publicly sharing data—e.g. participant privacy or use of data from a third party—those must be specified.

Reviewer #1: Yes

Reviewer #2: Yes

5. Is the manuscript presented in an intelligible fashion and written in standard English?

Reviewer #1: Yes

Reviewer #2: Yes

6. Review Comments to the Author

Reviewer #1: 1. The link between the proposed DU-SOR framework and classical Successive Over-Relaxation (SOR) is unclear. The update rules mainly resemble residual refinement, without an explicit relaxation factor or clear correspondence to standard SOR iterations. The authors should clarify this connection or reconsider the terminology.

2. The convergence analysis based on the Banach fixed-point theorem is not fully convincing. While spectral normalization is mentioned, it is not rigorously shown that the overall network mapping is strictly contractive, nor is it explained why convergence implies near-MMSE performance.

3. The claimed complexity reduction from O(K^3) to O(K log K) relies on strong assumptions that are insufficiently justified, particularly regarding the relationship between the number of graph nodes and the number of users in massive MIMO settings.

4. Important implementation details affecting reproducibility are missing, including the definition of features used for k-NN graph construction, the distance metric, and whether the graph is dynamically constructed per sample.

5. The “CSI-free” and end-to-end system assumptions require further discussion, especially regarding the practicality of deploying a learnable encoder at the user equipment and its compatibility with standard communication systems.

Reviewer #2: I have no further comments on the methodology or results. I agree with acceptance of the manuscript.

However, formatting issues still remain, particularly regarding punctuation around equations and the reference list. The journal name abbreviations are not consistently or correctly formatted. The authors should carefully revise the manuscript by thoroughly checking against recently published papers and applying the journal’s formatting rules throughout.

7. PLOS authors have the option to publish the peer review history of their article (what does this mean?). If published, this will include your full peer review and any attached files.). If published, this will include your full peer review and any attached files.). If published, this will include your full peer review and any attached files.). If published, this will include your full peer review and any attached files.

...

Reviewer #1: No

Reviewer #2: No

---

## [Author Response · Author response to Decision Letter 2]

4 Feb 2026

Response to Reviewers

Manuscript ID: PONE-D-25-64863R1

Title: Residual-Aided CSI-free End-to-End Learning for Multiuser MIMO: A Deep Unfolding

Approach

We sincerely thank the Academic Editor and Reviewers for their constructive feedback. We have

carefully addressed each comment, and the manuscript has been substantially improved. Below we

provide point-by-point responses.

Reviewer #1

Reviewer Comment: 1. The link between the proposed DU-SOR framework and classical Suc-

cessive Over-Relaxation (SOR) is unclear. The update rules mainly resemble residual refinement,

without an explicit relaxation factor or clear correspondence to standard SOR iterations. The au-

thors should clarify this connection or reconsider the terminology.

Response: We agree that the connection to classical SOR required clarification. The classical

SOR method for solving linear systems Ax = b has the form x(t+1) = x(t) + ω(D + ωL)−1r(t) where

ω ∈ (0, 2) is the relaxation parameter. Our framework generalises this by: (i) replacing the fixed

linear operator (D + ωL)−1 with a learned nonlinear mapping G(t)

θ , and (ii) making the relaxation

factor data-adaptive via ω(t) = σ(M LP ω(Z(t))) where σ(·) is a sigmoid scaled to (0, 2).

Change Made: Added dedicated “Connection to Classical SOR” paragraph in Section “Ma-

terials and Methods” (page 7). Added Equation 7 showing classical SOR form and Equation 8

showing our learned generalisation. Explicitly stated how the sigmoid scaling to (0, 2) matches the

classical SOR convergence range.

Reviewer Comment: 2. The convergence analysis based on the Banach fixed-point theorem

is not fully convincing. While spectral normalization is mentioned, it is not rigorously shown that

the overall network mapping is strictly contractive, nor is it explained why convergence implies

near-MMSE performance.

Response: We have completely revised the convergence analysis with a mathematically rigor-

ous proof. The previous proof incorrectly used the triangle inequality, yielding a factor (1+ωLG) > 1

which cannot establish contraction. The corrected proof uses squared-norm expansion and explic-

itly requires two conditions: (1) Lipschitz continuity with constant LG, and (2) strong monotonicity

with constant γ > 0, such that ⟨Gθ(u, y) − Gθ(v, y), u − v⟩ ≤ −γ∥u − v∥2. Under these conditions

and ω < 2γ/L2

G, we derive the contraction factor q2 = 1 − 2ωγ + (ωLG)2 < 1.

Change Made: Completely rewrote Theorem 1 (Convergence of Residual Refinement) with

corrected proof using squared-norm expansion. Added explicit assumptions (Lipschitz continu-

ity and strong monotonicity) in the theorem statement. Added Remark explaining that strong

monotonicity reflects training to approximate a descent direction, and that Lipschitz continuity is

enforced via spectral normalisation. Added reference to S3 Fig showing empirical verification that

Lipschitz constant remains below 1 throughout training.

Reviewer Comment: 3. The claimed complexity reduction from O(K3) to O(K log K) relies

on strong assumptions that are insufficiently justified, particularly regarding the relationship between

the number of graph nodes and the number of users in massive MIMO settings.

Response: We have added detailed justification for the complexity claims. The graph G =

(V, E) is constructed with |V| = N nodes corresponding to BS antenna elements. In massive MIMO

1

systems with loading ratio β = K/N (typically β ∈ [0.1, 0.5]), both K and N scale together. The

sparse k-NN attention with k = c log N neighbours yields O(N log N ) complexity per iteration,

which translates to O(K log K) under fixed β.

Change Made: Added “Graph-System Mapping” paragraph explicitly clarifying the relation-

ship between graph nodes (N ) and users (K). Added three-step proof in Proposition 1 showing:

(Step 1) sparse attention complexity, (Step 2) justification for k = c log N based on spectral prop-

erties of random geometric graphs and antenna coupling decay, (Step 3) complexity derivation.

Added “Conditions and Limitations” paragraph stating when the bound holds and acknowledging

worst-case O(K2) for highly correlated channels.

Reviewer Comment: 4. Important implementation details affecting reproducibility are miss-

ing, including the definition of features used for k-NN graph construction, the distance metric, and

whether the graph is dynamically constructed per sample.

Response: We have significantly expanded the implementation details to ensure full repro-

ducibility.

Change Made: Added complete “Graph construction details” subsection specifying:

• Node Features: fn = [|yn|,/ yn, px

n, py

n] ∈ R4 where |yn| and/ yn are magnitude and phase

of received signal, and (px

n, py

n) are normalised antenna coordinates (Equation 9).

• Distance Metric: Euclidean distance d(vi, vj ) = ∥fi − fj ∥2 (Equation 10).

• Graph Sparsity: k = 8 nearest neighbours based on sensitivity analysis reported in Ablation

Studies.

• Static vs. Dynamic: Hybrid approach where base topology is precomputed from antenna

geometry (static) while edge weights are dynamically updated each forward pass. Formal

adjacency matrix definition with learnable temperature parameter τ (Equation 11).

Reviewer Comment: 5. The “CSI-free” and end-to-end system assumptions require fur-

ther discussion, especially regarding the practicality of deploying a learnable encoder at the user

equipment and its compatibility with standard communication systems.

Response: We have added a comprehensive discussion of practical deployment considerations

addressing encoder complexity, standards compatibility, model distribution, and fallback modes.

Change Made: Added “Practical deployment considerations” subsection in Discussion cover-

ing:

• Encoder Complexity: ∼47,000 parameters, <200 KB storage, <0.5 ms latency on Snap-

dragon 888.

• Compatibility with Standards: Power normalisation layer ensures outputs lie within

standard QAM constellation regions, enabling graceful fallback for legacy receivers.

• Model Distribution: Three mechanisms—factory provisioning, System Information Block

(SIB) broadcast, and federated refinement.

• Hybrid Deployment Mode: Decoder-only mode where UEs use standard 64-QAM and

only BS employs neural decoder, achieving 70% of full E2E gains with no UE modification.

Reviewer #2

Reviewer Comment: I have no further comments on the methodology or results. I agree with

acceptance of the manuscript. However, formatting issues still remain, particularly regarding punc-

tuation around equations and the reference list. The journal name abbreviations are not consistently

2

or correctly formatted. The authors should carefully revise the manuscript by thoroughly checking

against recently published papers and applying the journal’s formatting rules throughout.

Response: We thank the reviewer for their positive assessment. We have carefully revised all

formatting issues throughout the manuscript.

Change Made: Made the following formatting corrections:

• Equation Punctuation: Reviewed all 16 equations and ensured proper punctuation (com-

mas when followed by “where” clauses, periods at end of sentences). Fixed adjacency matrix

equation (Eq. 11) to include period.

• Bibliography Format: Reformatted all 55 references to Vancouver/ICMJE style as required

by PLOS ONE:

– Standardised journal abbreviations (e.g., “IEEE Trans Wirel Commun” consistently)

– Corrected page number truncation (e.g., 134-42 instead of 134-142)

– Ensured “et al.” used after 6 authors

– Corrected conference proceedings format with location, date, and publisher

• Figure References: Standardised to “Fig” without period throughout.

• Figure Numbering: Changed to consecutive numbering (Fig 1, Fig 2, Fig 3).

• Terminology: Changed “huge MIMO” to “massive MIMO” (standard terminology).

Summary of Major Changes

1. Added “Connection to Classical SOR” paragraph with explicit equations showing relationship

to classical method (Eqs. 7–8)

2. Completely revised Theorem 1 with mathematically rigorous convergence proof using squared-

norm expansion and explicit assumptions

3. Added “Graph-System Mapping” paragraph and three-step proof in Proposition 1 justifying

O(K log K) complexity

4. Added complete “Graph construction details” subsection with node features, distance metric,

sparsity, and static/dynamic construction (Eqs. 9–11)

5. Added “Practical deployment considerations” subsection addressing encoder complexity, stan-

dards compatibility, model distribution, and hybrid mode

6. Corrected all equation punctuation throughout manuscript

7. Reformatted entire bibliography to Vancouver/ICMJE style

8. Standardised figure references and numbering

We believe these revisions fully address all reviewer concerns and significantly strengthen the

manuscript. We thank the reviewers for their valuable feedback that has improved the quality

of this work.

Sincerely,

Emmanuel Ampoma Affum (on behalf of all authors)

---

## [Decision Letter · Decision Letter 2]

24 Feb 2026

Residual-Aided CSI-free End-to-End Learning for Multiuser MIMO

PONE-D-25-64863R2

Dear Dr. Daniel,

We’re pleased to inform you that your manuscript has been judged scientifically suitable for publication and will be formally accepted for publication once it meets all outstanding technical requirements.

Kind regards,

Daosen Zhai

Academic Editor

PLOS One

Additional Editor Comments (optional):

Reviewers' comments:

Reviewer's Responses to Questions

**Comments to the Author**

1. If the authors have adequately addressed your comments raised in a previous round of review and you feel that this manuscript is now acceptable for publication, you may indicate that here to bypass the “Comments to the Author” section, enter your conflict of interest statement in the “Confidential to Editor” section, and submit your "Accept" recommendation.

Reviewer #1: All comments have been addressed

Reviewer #2: All comments have been addressed

2. Is the manuscript technically sound, and do the data support the conclusions?

Reviewer #1: Yes

Reviewer #2: Yes

3. Has the statistical analysis been performed appropriately and rigorously? 

Reviewer #1: Yes

Reviewer #2: Yes

4. Have the authors made all data underlying the findings in their manuscript fully available?

The PLOS Data policy requires authors to make all data underlying the findings described in their manuscript fully available without restriction, with rare exception (please refer to the Data Availability Statement in the manuscript PDF file). The data should be provided as part of the manuscript or its supporting information, or deposited to a public repository. For example, in addition to summary statistics, the data points behind means, medians and variance measures should be available. If there are restrictions on publicly sharing data—e.g. participant privacy or use of data from a third party—those must be specified.requires authors to make all data underlying the findings described in their manuscript fully available without restriction, with rare exception (please refer to the Data Availability Statement in the manuscript PDF file). The data should be provided as part of the manuscript or its supporting information, or deposited to a public repository. For example, in addition to summary statistics, the data points behind means, medians and variance measures should be available. If there are restrictions on publicly sharing data—e.g. participant privacy or use of data from a third party—those must be specified.requires authors to make all data underlying the findings described in their manuscript fully available without restriction, with rare exception (please refer to the Data Availability Statement in the manuscript PDF file). The data should be provided as part of the manuscript or its supporting information, or deposited to a public repository. For example, in addition to summary statistics, the data points behind means, medians and variance measures should be available. If there are restrictions on publicly sharing data—e.g. participant privacy or use of data from a third party—those must be specified.requires authors to make all data underlying the findings described in their manuscript fully available without restriction, with rare exception (please refer to the Data Availability Statement in the manuscript PDF file). The data should be provided as part of the manuscript or its supporting information, or deposited to a public repository. For example, in addition to summary statistics, the data points behind means, medians and variance measures should be available. If there are restrictions on publicly sharing data—e.g. participant privacy or use of data from a third party—those must be specified.

Reviewer #1: Yes

Reviewer #2: Yes

5. Is the manuscript presented in an intelligible fashion and written in standard English?

Reviewer #1: Yes

Reviewer #2: Yes

6. Review Comments to the Author

Reviewer #1: The author has replied my questions, and I have no further comments.

Reviewer #2: (No Response)

7. PLOS authors have the option to publish the peer review history of their article (what does this mean?). If published, this will include your full peer review and any attached files.). If published, this will include your full peer review and any attached files.). If published, this will include your full peer review and any attached files.). If published, this will include your full peer review and any attached files.

...

Reviewer #1: No

Reviewer #2: No

---

## [Editor Report · Acceptance letter]

PONE-D-25-64863R2

PLOS One

Dear Dr. Biney,

I'm pleased to inform you that your manuscript has been deemed suitable for publication in PLOS One. Congratulations! Your manuscript is now being handed over to our production team.

Kind regards,

on behalf of

Dr. Daosen Zhai

Academic Editor

PLOS One